# Can Humanized Immune System Mouse and Rat Models Accelerate the Development of Cytomegalovirus-Based Vaccines Against Infectious Diseases and Cancers?

**DOI:** 10.3390/ijms26073082

**Published:** 2025-03-27

**Authors:** Kaci Craft, Athina Amanor, Ian Barnett, Clarke Donaldson, Ignacio Anegon, Srinivas Madduri, Qiyi Tang, Moses T. Bility

**Affiliations:** 1Department of Microbiology, Howard University College of Medicine, Washington, DC 20059, USA; kaci.craft@bison.howard.edu (K.C.); athina.amanor@bison.howard.edu (A.A.); ian.barnett@bison.howard.edu (I.B.); clarke.donaldson@bison.howard.edu (C.D.); qiyi.tang@howard.edu (Q.T.); 2Nantes Université, INSERM, Center for Research in Transplantation and Translational Immunology, UMR 1064, F-44000 Nantes, France; ignacio.anegon@univ-nantes.fr; 3Bioengineering and Neuroregeneration Laboratory, Department of Surgery, University of Geneva, 1211 Geneva, Switzerland; srinivas.madduri@unige.ch

**Keywords:** humanized immune system and rats, HIV vaccines, HCMV-based vaccines, HIV/AIDS-animal models, human cancer xenograft models

## Abstract

Over the past three decades, immunodeficient mouse models carrying human immune cells, with or without human lymphoid tissues, termed humanized immune system (HIS) rodent models, have been developed to recapitulate the human immune system and associated immune responses. HIS mouse models have successfully modeled many human-restricted viral infections, including those caused by human cytomegalovirus (HCMV) and human immunodeficiency virus (HIV). HIS mouse models have also been used to model human cancer immunobiology, which exhibits differences from murine cancers in traditional mouse models. Variants of HIS mouse models that carry human liver cells, lung tissue, skin tissue, or human patient-derived tumor xenografts and human hematopoietic stem cells-derived-human immune cells with or without lymphoid tissue xenografts have been developed to probe human immune responses to infections and human tumors. HCMV-based vaccines are human-restricted, which poses limitations for mechanistic and efficacy studies using traditional animal models. The HCMV-based vaccine approach is a promising vaccine strategy as it induces robust effector memory T cell responses that may be critical in preventing and rapidly controlling persistent viral infections and cancers. Here, we review novel HIS mouse models with robust human immune cell development and primary and secondary lymphoid tissues that could address many of the limitations of HIS mice in their use as animal models for HCMV-based vaccine research. We also reviewed novel HIS rat models, which could allow long-term (greater than one year) vaccinology studies and better recapitulate human pathophysiology. Translating laboratory research findings to clinical application is a significant bottleneck in vaccine development; HIS rodents and related variants that more accurately model human immunology and diseases could increase the translatability of research findings.

## 1. Introduction

Mouse models are invaluable pre-clinical platforms for developing vaccines against infectious diseases and cancer [1,2,3,4,5,6,7,8,9,10,11,12,13,14]. However, gaps exist in translating the findings from vaccine studies in mouse models to the clinic [15]. Significant gaps include the differences in immune response between human and murine immune cells [16,17] and biological differences between many infectious diseases and cancers in mice and humans [15]. Additionally, many viral-vector vaccines, such as cytomegalovirus (CMV)-based vaccines, are species-restricted, thus limiting the evaluation of the safety and efficacy of human-specific viral vector vaccines [16,17]. The virology of human CMV-based vaccines has predominately been elucidated using human cell lines; however, immunology studies are limited [17]. Surrogate rhesus-CMV viral vectors have been used to study CMV-based vaccines to elucidate the determinants of immunity in humans [18]. However, significant differences exist between human CMV and rhesus CMV and between human immune cells and rhesus immune cells [19]. To bridge this research gap, mice with human immune cells, with or without human lymphoid tissues, termed human immune system (HIS)-mouse models and related variants, have been used in biomedical research, particularly in immunology and virology [20,21,22,23,24,25]. These models, which incorporate human hematopoietic stem cells, peripheral blood cells, lymphoid tissues, and other tissues, provide a unique platform for studying human-specific biological processes in a controlled, in vivo environment [5,6,7,8,9,10,11,12,13,14]. By enabling researchers to mimic human-specific biological responses, these models help bridge the gap between basic research and clinical applications, allowing for a deeper understanding of disease mechanisms, immune responses, and the development of targeted therapies [20,21,22,23,24,25,26,27,28]. These advances in modeling human biology in HIS mice have also been extended to other rodents, namely HIS rats [29,30]. HIS rat models address current HIS mouse models’ lifespan and size limitations and more accurately mimic human physiology [29,30,31]. Here, we discuss the advances in HIS rodent models and their possible applications in accelerating the development of cytomegalovirus-based vaccines against infectious diseases and cancers.

## 2. Humanized Immune System (HIS) Rodent Models and Related Variants

Many pathogens that infect humans are restricted to humans, thus limiting the utility of traditional rodent models for infection, immunology, and pathogenesis studies [1,2]. Furthermore, human cancer biology exhibits differences from the cancer biology in traditional rodent models [3,4,15]. Over the past few decades, investigators have developed immunodeficient mouse models via genetic engineering to support the engraftment and development of human cells and tissues in severely immunodeficient mouse models [1,2]. These immunodeficient mouse models typically involve deleterious mutations or deletions of the recombination-activating genes (RAGs) and the Interleukin 2 Receptor Subunit Gamma (IL2RG) gene [1,2]. These genetic deficiencies result in T, B, and NK cell deficiencies [29,32]. Additionally, the mouse models derived from the NOD strain have a deficiency in macrophage and dendritic cell functions, which is mediated via spontaneous deleterious mutation in murine signal regulatory protein alpha (murine SIRPa) [29,32]. This murine-myeloid cell deficiency can also be introduced via the knock-in expression of human signal regulatory protein alpha (human SIRPa) [29,32]. These immunodeficiencies prevent the host-versus-graft immune response, provide tissue space for the human cells, and allow the engrafted cells/tissues to develop [17,29,31,33] (Figure 1). These genetic engineering advances have recently been extended to rat models [34], yielding similar results in the engraftment and development of human immune cells in immunodeficient rats [29,31] (Figure 1). Additionally, the lymphoid cell subsets in the blood of HIS rats are like the ones in human blood, with a predominance of T cells compared to B cells [29].

Additional modifications of HIS mouse models have also included the introduction of inducible cell death in other murine tissues, such as the liver, to promote the engraftment and development of human liver cells [21,23].

Several HIS mouse models have also introduced human cytokines (albeit at non-physiological levels) to improve the development of specific immune lineages [35,36]. Future advances that replace rodent hormones, growth factors, and cytokines in the lymphoid tissue stroma or other tissue stroma with the human counterpart at physiological levels may be necessary to mimic human cell/tissue development in humanized rodents accurately [37,38]. 

Widely used HIS mouse models incorporate human CD34+ hematopoietic stem cells-derived de novo human immune cells, with or without autologous human lymphoid tissues (primary—thymus, and secondary—spleen) [25,39] (Figure 1).

These HIS mouse models have emerged as invaluable tools in biomedical research, particularly in immunology and virology [18,20,23,25,39,40]. These models and related variants provide a unique platform for studying human-specific biological processes in a controlled in vivo environment [40,41]. By enabling researchers to recapitulate human biological responses, these models help bridge the gap between tissue culture studies and clinical studies, allowing for a deeper understanding of disease mechanisms and the development of targeted therapies [40,41,42].

The human immune system is more complex than tissue culture models and significantly different from murine immune systems [43]. HIS mouse models and related variants address this disparity by engrafting human immune cells or tissues into immunodeficient mice, creating a more relevant biological context for studying human immune responses [1,2,3,4,5,6,7,8,9,10,11,12,13,14]. HIS mouse models have proven to be instrumental in understanding human immune signaling pathways, such as those involved in T cell activation [44,45,46,47], B cell differentiation [48,49,50], and cytokine signaling [51,52]. Researchers have utilized HIS-mouse models to tackle various immune-related research questions involving infection, autoimmune diseases, allergies, and cancer [4,10,16,53,54,55,56,57]. For instance, studies have shown that HIS mice can recapitulate human immune responses to HIV, which is critical for developing vaccines and therapeutics for HIV cure and prevention [9,25]. HIS mouse model variants incorporating other tissues (such as liver, lung, etc.) also enable analysis of human immune cells-virus interaction, providing insights into immune signaling pathways that are vital for effective immune responses [40,58].

Furthermore, HIS mouse models are utilized to investigate the efficacy and safety of immunotherapies against human tumor xenografts [4,55,59,60]. This has led to significant advancements in manipulating immune responses to neoplastic diseases, particularly cancers that exploit immune pathways to evade immunotherapy interventions [4,55,59,60].

## 3. Modeling Pathogen Infections and Human Cancer in HIS Rodent Models and Related Variants

The HIS mouse models have been utilized to study the infection, immune response, and pathogenesis of a wide range of pathogens, including human immunodeficiency virus (HIV) and human cytomegalovirus (HCMV) [16,24,25,39] (Figure 2). HIS mouse and rat models have also been modified to engraft and develop non-hematopoietic lineage cells/tissues, including human lung tissue [40], human skin tissue [31], and human liver cells [23,61]. These modified HIS mouse models enable in vivo mechanistic studies on the immunology and pathogenesis of human pathogens that target non-hematopoietic lineage cells/tissues (i.e., Staphylococcus aureus—skin [31], hepatitis viruses—hepatocytes [23,58,61], coronavirus—lungs) [40]. The future engraftment and development of other non-hematopoietic lineage cells/tissues, such as peripheral nerves, central nervous tissue, muscles, and dental tissues in humanized rat models, will expand the range of pathogens that can be studied. In a similar vein, a novel use of immunodeficient rodents has been the in vivo development of organoids generated from pluripotent stem cells, such as for intestinal organoids in immunodeficient rats [62] and different organoids in immunodeficient mice and rats [63,64,65].

HIS mouse models and related variants have had the most impact on the field of virology [2]. HIS mouse models and related variants are essential for studying the virology and immunobiology of human-restricted viruses [2]. Traditional rodent models often fail to replicate human viral infectious diseases due to species-specific differences in cellular receptors for viral infection and immune responses [2]. HIS mice and related variants provide a relevant system for investigating the mechanisms of viral entry, replication, and immune evasion [1]. These models have been particularly valuable in studying emerging viruses such as HIV [66], the Zika virus [67], and the coronavirus [40]. Researchers can observe viral infection dynamics in a context that recapitulates human pathophysiology and immune responses by incorporating human tissues that express specific receptors necessary for viral entry [40,57]. HIS mouse models and related variants have significantly advanced the understanding of HIV virology and immunology, as well as the effects of co-morbidities [26,27,68]. HIS mouse models have been used to investigate HIV infection dynamics, virulence factors, and anti-viral response, identifying key viral pathways and host factors that enable infection and pathogenesis [24,26,27,68,69,70,71]. 

HIS mouse models and related variants are crucial for evaluating antiviral therapies and vaccine candidates [24,57,69,70,71] (Figure 2). Researchers have employed HIS mice and related variants to assess the efficacy of anti-viral therapeutics [72] and Immunotherapeutics [73,74], observing the efficacy against viral load and pathogenesis as well as the associated immune responses within a human-like tissue microenvironment. This capability of HIS mouse models accelerates the development of effective treatments and vaccines. HIS mouse models have significantly impacted the development of anti-HIV drugs and immunotherapies [1,24,57]. HIS mouse models have also provided the ideal in vivo platform for mechanistic studies on drugs and therapeutic antibody [75,76,77] efficacy and resistance in HIV infection.

In addition to modeling pathogen infection, HIS mice have also been used for investigating immune responses against a myriad of human tumors (Figure 2) [4,56,78]. It is well established that chemically and genetically induced rodent tumor models have limited inflammation [79]; this contrasts with human tumors primarily driven by inflammation [3,4,41,56,59,78]. HIS mouse models support the development of human tumor xenografts and the associated inflammatory tumor microenvironment and recapitulate tumor immunobiology [4,41,59,78]. Advances have also been made in developing human tumor stroma to mimic the tumor microenvironment and the associated stromal signaling that modulates immune responses to human cancers [41]. Similarly, immunodeficient rats have been used to analyze carcinogenesis and responses to treatments of diverse human cancers, such as leukemia, gliomas, and mammary and lung cancers [80,81,82,83,84,85]. These features make HIS rodents ideal for studying immunotherapies and vaccines against human cancers (Figure 2). Future advances in modeling the human tumor microenvironment and associated tumor immunobiology in HIS rodents will advance cancer vaccine efficacy studies. 

## 4. HIS-Rodent Models of HCMV Infection and Immune Response

Over the past 30 years, several studies have demonstrated that HIS mouse models support HCMV infection, pathogenesis, and immune response [86,87,88,89,90,91,92,93,94,95,96]. HIS mouse models are the ideal small animal models for in vivo mechanistic studies of HCMV, a human-specific beta-herpesvirus that infects myeloid progenitor cells [86,87,88,89,91,92,93,94,95]. HCMV also infects a myriad of other cell types, including hematopoietic and non-hematopoietic lineage cells [93,94,95,96]. Myeloid cells are the most important hematopoietic lineage cell type with respect to HCMV replication, latency, reactivation, and persistence. Non-hematopoietic lineage cells, such as stromal cells, endothelial cells, epithelial cells, fibroblasts, neuronal cells, and smooth muscle cells, also support HCMV infection. 

Most HIS mouse models have some degree of myeloid cell reconstitution, thus rendering them susceptible to HCMV infection. Additionally, HIS mice reconstituted with human lymphoid tissues are also susceptible to HCMV infection, as they contain both hematopoietic lineage cell targets (human lymphoid tissue macrophages) and non-hematopoietic-lineage cell targets (human lymphoid tissue stromal cells, endothelial cells, epithelial cells, and fibroblasts) [93,94,95,96,97]. Indeed, the first studies demonstrating HCMV infection in HIS mice were conducted in an immunodeficient mouse model engrafted with human lymphoid tissue (human fetal thymus and fetal liver) in the kidney capsule, termed SCID-hu Thy/Liv humanized mice, a first-generation-like HIS mouse model [93,94,95,96,97]. Several studies demonstrated that the engrafted human thymus in the SCID-hu Thy/Liv humanized mouse model supported HCMV (HCMV-Toledo, AD169, and Towne strains) infection and replication [92,97]. Additionally, transplantation of granulocyte-colony stimulating factor (G-CSF)-mobilized stem cells from HCMV seropositive donors into severely immunodeficient mice (second-generation HIS mice) supports HCMV replication. 

The second-generation HIS mouse models are reconstituted systematically with monocytes, macrophages, and B and T cells following transplantation of human CD34+ hematopoietic stem cells; albeit the myeloid cell reconstitution is suboptimal. These second-generation HIS mouse models are not only susceptible to HCMV infection but also support human T and B cells’ antiviral immune responses; however, those responses are limited. Several studies using second-generation HIS mice demonstrated that the engrafted human cells supported HCMV infection and replication [87,88]. Furthermore, transplanting G-CSF-mobilized peripheral blood-hematopoietic stem cells from HCMV-seropositive donors into immunodeficient mice results in HCMV replication [87,88]. 

To significantly improve adaptive immune cell responses, third-generation HIS-mouse models were developed via transplanting human lymphoid tissues along with autologous human CD34+ hematopoietic stem cells [2,39,98]. These third-generation HIS mouse models enable improved T cell education in the human thymus and the systemic reconstitution of a myriad of human immune cells, including myeloid lineage cells [2,39]. The incorporation of both human thymus and human spleen in HIS mice further improves the immune system [39]. Several studies have demonstrated HCMV replication and the associated human adaptive immune response in third-generation HIS mouse models; however, the secondary lymphoid tissues in those animals are suboptimal [16,17,40,89,99]. Improvements in the secondary lymphoid tissues in HIS mice and the incorporation of non-hematopoietic cell targets (i.e., human lung tissues) could enhance HCMV infection and improve immune responses. 

## 5. The Biology of HCMV-Based Viral Vectored Vaccines

Viral vectors are modified viruses specifically engineered to deliver foreign genes into host cells efficiently, leveraging viruses’ natural gene transfer capabilities and immune-stimulatory properties. The most used viruses include Adenovirus (Adv), Adeno-associated virus (AAV), lentivirus, retrovirus, herpes simplex virus (HSV), vaccinia virus, baculovirus, Sendai virus, and Rabies virus [100,101,102,103,104,105,106,107]. These viral vectors have unique properties that make them suitable for different applications in gene therapy, vaccine development, and genetic research. However, AAV, lentivirus, and retrovirus possess genomic integration capabilities that restrict their use in genetic diseases and cancers. Adv and HSV vectors are limited because of their existing immunogenicity. RNA virus vectors have a small packaging capacity for foreign genes. Therefore, the development of better viral vectors is necessary. Human cytomegalovirus (HCMV) has unique features that address many of the current limitations of viral vector vaccines.

HCMV, also known as human herpesvirus 5 (HHV-5), is a widespread beta-herpesvirus belonging to the herpesvirus family [108,109]. It is characterized by a large, double-stranded DNA genome (~235 kbp) that encodes over 200 genes [110]. HCMV has garnered significant attention as a promising viral vector for gene delivery due to its unique biological features and engineering versatility. The HCMV genome has been cloned into a bacterial artificial chromosome (BAC), enabling precise genetic modifications to express foreign genes [9,12]. Its large genomic capacity supports the inclusion of multiple foreign genes, making it particularly valuable for developing polyvalent vaccines. HCMV has a high global seroprevalence [109]. While it is typically asymptomatic in immunocompetent individuals, it can cause severe complications in immunocompromised individuals, neonates, and organ transplant recipients [110]. HCMV establishes lifelong latency in hematopoietic cells, with reactivation occurring under conditions of immunosuppression [109]. The virus spreads through bodily fluids, such as saliva, blood, and breast milk, as well as via organ transplantation. These properties underscore the clinical significance of HCMV and highlight its potential as a versatile platform for developing vaccines and therapeutic vectors. Additionally, CMV can bypass pre-existing immunity and elicit strong T-cell responses [110]. Furthermore, emerging evidence suggests that HCMV produces circular RNA, which could be leveraged in the development of HCMV-based circular RNA vaccines [111]. 

Extensive research has demonstrated that CMV-based vectors are versatile tools and capable of expressing genes related to a wide range of pathogens, including malaria [112], tuberculosis [15], Ebola [113,114], HIV/SIV [19,115], COVID-19 [116], influenza [117] and cancer [118]. Importantly, CMV does not integrate viral genomic DNA into the host genome [119,120]. Moreover, single-cycle HCMV vectors, created by deleting an essential gene, can be used in immunocompromised populations due to their inability to replicate [120]. These vectors are highly efficient gene transfer agents, making them valuable in genetic research and therapeutic interventions [120]. 

## 6. Vaccinology Considerations for HCMV-Based Viral Vectored Vaccines

HCMV can induce lifelong immunity because of its two characteristics. First, HCMV has a unique ability to establish a lifelong latent infection within the host, meaning the virus remains present in the body even when not actively replicating. This persistent infection allows for the continuous expression of viral antigens, which constantly stimulate the immune system to maintain surveillance against the virus [121]. Additionally, HCMV can periodically reactivate from latency, leading to a renewed presentation of viral antigens to the immune system [121]. This repeated antigen presentation during reactivation cycles significantly contributes to expanding and maintaining a robust population of memory T cells specifically targeted against HCMV [121]. The ability of HCMV to induce robust and lifelong immunity makes it an ideal viral vector for vaccines against persistent infections and cancer [118,121].

Several studies have demonstrated that CMV-based vaccines can induce robust cellular immunity by eliciting broad, robust, and polyfunctional CD8+ and CD4+ T-cell responses, including non-canonical MHC-E-restricted CD8+ T cells (less prone to viral immune escape), and can activate strong NK cells due to CMV-mediated stress signals [18,19,107,122,123,124,125]. Unlike typical MHC Class I-restricted CD8+ T cells, CMV-based vaccines can also stimulate CD8+ T cells that recognize viral peptides presented by the non-classical MHC-E molecule [19,107,122,123,124,125]. This unique feature can provide a more resilient vaccine-induced immune response against viral escape mechanisms [122,125,126]. CMV-vectored antigens are also efficiently presented via both the MHC-I and MHC-II pathways, ensuring the activation of both CD8+ and CD4+ T cells [122,126]. Additionally, CMV modulates innate responses via the toll-like receptor (TLR) and interferon pathways, amplifying vaccine efficacy [127]. The safety of viral vectors is critical for clinical use as vaccines; live HCMV vectors are safe in immunocompetent hosts. Attenuated HCMV strains and engineered replication-deficient vectors are also safe in immunocompetent individuals [19]. Notably, the killed or once-cycle replicative HCMV vector is safe for immunocompromised individuals, providing alternatives for at-risk populations [19,121].

The ability to carry large inserts to produce multiple antigens is essential for viral vectors [120]. HCMV has a large genome capacity; thus, it can accommodate large or multiple transgene inserts (~30–40 kb capacity) without compromising replication, allowing the expression of multiple antigens or additional immune-enhancing genes, such as IL-15 [118]. It should be noted that the large capacity needs BAC-mediated bioengineering. Additionally, HCMV can efficiently infect mucosal tissues, a primary entry site for many pathogens, enabling a strong mucosal immune response (critical for HIV, influenza, etc.) against desired antigens [128,129]. Furthermore, HCMV can infect a wide range of cell types, including those found in the nasal mucosa [128]. It can also be detected in bodily fluids secreted from mucosal surfaces, such as saliva, urine, cervical fluid, semen, and breast milk [128,130]. 

Despite the advantages, the HCMV vector faces some challenges. First, latent/persistent infection in immunocompromised hosts could pose risks [107]. Alternatively, a single-cycle version of the HCMV vector used for immunocompromised individuals should be generated along with the replicative version of the HCMV vector that is used for immunocompetent individuals. Secondly, human CMV (HCMV) and animal CMVs are highly species-specific, infecting only their natural host [89]. For example, HCMV infects only humans, and rhesus CMV (RhCMV) only infects macaques and closely related species [89,122]. This brings a bottleneck in designing the animal study model for HCMV-vectored vaccines. Thus far, humanized mice are the closest model for studying the efficiency of HCMV-vector-induced immune reactivity [86,98]. In addition, HCMV evades host immunity by inhibiting antigen presentation via MHC-I and MHC-II, modulating cytokine responses, and/or downregulating NK cell activation via UL16-binding proteins [107]. Fortunately, we can use a BAC-mediated bioengineering method to enable HCMV to avoid evading the immune system [107,122]. Additionally, vaccine strains must be attenuated to prevent disease in immunocompromised individuals; strategies include deletions of essential replication genes (e.g., UL128-131 locus) [19,107]. Lastly, most of the human population has pre-existing immunity; however, studies using RhCMV vaccine strains have demonstrated that pre-existing seropositivity does not affect vaccine efficacy [19,125]. Ongoing clinical studies using HCMV-based vaccine strains comparable to RhCMV-based vaccine strains will address concerns regarding the effect of pre-existing seropositivity on vaccine efficacy in humans [131].

As mentioned above, CMV infection is strictly limited by species-specificity, HCMV-vector studies have been limited, and the FDA has approved no HCMV-vectored vaccines [131]. Other animal CMV-vectored vaccines have been under examination, providing insights into further studies for HCMV. The following vaccines are currently under investigation: RhCMV-vectored SIV vaccine, MCMV-vectored influenza virus vaccine, RhCMV-vectored Ebola virus vaccine, RhCMV-vectored malaria vaccine, MCMV-vectored RSV vaccines, and HSV-1 vaccines [18,107,132,133,134]. CMV vectors can be used to develop anti-cancer vaccines, such as the RhCMV-vectored vaccine expressing tumor antigens [124]. The detailed CMV-vectored vaccines were summarized previously [108].

## 7. Application of HIS-Rodent Models in Accelerating the Development of HCMV-Based Viral Vector Vaccines Against Infectious Diseases and Cancers

HCMV-based vaccines could induce robust and persistent immunity, which may provide the ideal vaccine platform for preventing and treating persistent infections and cancers [18,107,129]. Like HCMV, HCMV-based vaccines infect human myeloid cells, fibroblast, and lymphoid cells, each playing a unique role in infection and the associated immunity [107,123]. Monocytes, macrophages, and dendritic cells help HCMV spread through the blood and serve as long-term reservoirs [123]. In addition, fibroblast cells support viral replication and local tissue spread, making them essential for studying how HCMV and HCMV-based vaccines multiply in different immune system compartments, including secondary lymphoid tissues [19]. Live-attenuated HCMV-based vaccines against persistent infections (such as HIV), and cancers are in varying stages of development; however, these vaccines are primarily investigated using surrogate-animal CMV vaccines [18,107,129]. Recent advances in HIS rodents provide a platform for investigating the virology and immunobiology of HCMV-based vaccines against persistent human pathogens (such as HIV) and cancers [17,98]. Furthermore, these advances in humanized rodent models also enable mechanistic studies on HCMV-based vaccines that cannot be investigated in human studies due to the inability to obtain significant organs, invasive sampling, or human-pathogen challenge [89] (Figure 2).

Recent studies demonstrated that a third-generation HIS mouse model (carrying hematopoietic stem cell derived-de novo immune cells and autologous human thymus and spleen), termed BLTS-humanized mice could control replication-competent, live-attenuated HIV (aviremic infection) [25]. The viremic control of live-attenuated HIV in the BLTS-humanized mice contrasted with the robust replication of the live-attenuated viral stain in first-generation HIS mice with predominantly CD4+ T cell targets [25]. This finding demonstrated for the first time that a functional human immune system can be developed in HIS mouse models to control a viral infection [25]. An analysis of the immune response against live-attenuated HIV (Nef-defective HIV) demonstrated a robust anti-viral innate immune response and adaptive immune response [25].

Interestingly, there were several overlaps between the immune signaling associated with the viremic control of live-attenuated HIV (Nef-defective HIV) in the BLTS-humanized mouse model and RhCMV-based SIV vaccine-induced immunity in the rhesus macaque-surrogate model of HIV/AIDS [18,25]. IL-15, NK cell, and T cell signaling were significant features in the viremic control of live-attenuated HIV (Nef-defective HIV) in the BLTS-humanized mice and RhCMV-based SIV vaccine-induced immunity in rhesus macaques challenged with pathogenic SIV [18,25]. Of significance, the critical immune signature in the BLTS-humanized mice was of human origin, thus demonstrating in a laboratory setting the importance of those immune signaling pathways in HIV immunity [25]. Lymphoid tissue-immune cells and associated stromal cells, including fibroblasts, are crucial for mounting immune responses, and their role is evident in the BLTS-humanized mouse model, which successfully controlled HIV through IL-15, NK cells, and T cell activation [25]. These findings demonstrate that the BLTS-humanized mouse model can be used for mechanistic studies of the virology and immunobiology of HCMV-based vaccines, as the model is capable of mediating critical features of CMV-based vaccine immune signaling [19,25] (Figure 3). Furthermore, the BLTS-humanized mouse model can support HIV challenge studies [25] and human tumor development [4], thus enabling vaccine efficacy evaluation (Figure 3). 

A significant limitation of HIS mouse models that limits their application in vaccine studies against persistent human infections (such as HIV) and cancers is the relatively short lifespan (<one year) [2]. The approximately three- to six-month experimental window for generating vaccine-induced immunity and challenging the animals with wild-type viruses or cancers constrains their use in vaccine efficacy and safety studies [2]. However, opportunities remain to apply HIS mouse models to accelerate the development of HCMV-based viral vector vaccines against human infectious diseases and cancers. Third-generation models like the BLTS-humanized mice can enable mechanistic studies of the immune response generated by an HCMV-based vaccine [25] (Figure 3). This includes applying mechanistic insights from animal CMV-based vaccine studies to optimize comparable HCMV-based vaccines [19] (Figure 3). The application of HIS mice could reduce the time needed for the iteration process and feedback loop between pre-clinical animal studies and clinical optimization for immunogenicity and safety [19] (Figure 3). Third-generation HIS mouse models can also provide information on the vaccine-induced immune response to wild-type viruses or tumor challenges (Figure 3). 

Currently, an HCMV-based HIV vaccine, namely VIR-1388, is under clinical investigation (HVTN 142) to assess its safety and immunogenicity [131]. The live-attenuated HCMV-vector HIV vaccine (VIR-1388) is based on the RhCMV-vector SIV vaccine [19,131]. VIR-1388 is currently in a Phase 1 clinical trial involving HIV-negative participants who have asymptomatic CMV. The primary focus of this clinical trial is to analyze VIR-1388’s safety and potential for initiating an HIV-specific immune response in individuals with asymptomatic CMV [131]. There are several limitations in translating the findings from RhCMV-based SIV vaccine studies to HCMV-based HIV vaccine clinical trials due to species differences in vector biology and immunobiology [19]. Additionally, limitations in sampling or populations (asymptomatic CMV+ individuals) impair the rapid development of HCMV-based vaccines [131]. These limitations could be mitigated using animal models that enable mechanistic studies of the virology and immunobiology of HCMV-vector vaccines [86] (Figure 3).

HIS rodent models are increasingly being employed to support research and assess the immunogenicity of HCMV-based vaccinations in an in vivo platform [86]. These immunocompromised rodent models carrying human immune cells and lymphoid tissues allow scientists to simulate human immunological responses to vaccination. Using these models, scientists can assess the kinetics of human immune response activation, correlations of human immunity, possible toxicity, and infectious disease and cancer immuno-regulatory processes before proceeding to human trials (Figure 3). These models also enable the rapid iteration of vaccine design (Figure 3). Integrating HIS rodent models and related variants with human clinical trials provides a synergistic approach to evaluating the potential of HCMV-based HIV vaccines (Figure 3). HIV vaccine studies offer an excellent opportunity to assess the application of HIS rodent models in HCMV-based vaccine development (Figure 3). Although an HCMV-based HIV vaccine is undergoing clinical investigation in humans for immunogenicity, the human studies are predominantly guided by findings from surrogate RhCMV-based SIV vaccines [19]. The lack of information from animal studies using HCMV-based HIV vaccines is a significant knowledge gap, as the virology of human CMV and rhesus CMV differ [19]. This knowledge gap can be bridged using HIS-rodent models (Figure 3). 

HIS-rodent models could also provide invaluable information on the tropism and expression of HCMV-viral vector genes and vaccine antigens, potentially mitigating safety issues [32] (Figure 3). For instance, HIS-mice played a significant role in delineating the in vivo mechanisms of HIV transmission, demonstrating that viral tropism in immune cells is mediated via CD4 and co-receptors, namely, CCR5 and CXCR4 [70,72,135,136,137]. Evaluating the efficacy and safety of HCMV-based HIV vaccines in HIS-rodents is worthwhile before embarking on clinical studies in humans to avoid catastrophic outcomes, such as the STEP HIV vaccine trial, which resulted in an increased susceptibility to HIV transmission [138,139]. HIS mice have also enabled the intricate modeling of various aspects of HIV transmission that mimic clinical conditions, including the use of clinically relevant HIV strains and co-infection with other typical human sexually transmitted diseases [57,140]. Knowledge of the impact of these biological processes is critical for designing vaccine efficacy and safety studies in humans; however, surrogate virus-animal models are incapable of addressing these concerns [19]. Similar concepts can also be applied to HCMV-based cancer vaccines, as recent advances in tumor modeling in HIS mice enable closing the translation gap between laboratory research and clinical efficacy and safety studies [41].

## 8. Future Directions for Immunobiology Modeling in HIS-Rodents and Related Variants

Future efforts to improve HCMV-based vaccine studies in HIS-rodents must address the issue of available experimental windows. Developing a third-generation, human-like immune system in HIS-rats could enable long-term studies (exceeding 2 years) for evaluating vaccine efficacy and safety following challenges with wild-type pathogens or tumors. Furthermore, the application of novel genome editing techniques to rats is rapidly generating many new models [34], and negating the “murine model—genetic manipulation advantage” [141] in biomedical research. New cytokine conditioning strategies (i.e., 17β-estradiol [48]) for improving human lymphoid development will likely be applied to immunodeficient rats, as has been done with mouse models. The larger size of rats compared to mice allows for robust longitudinal sampling, testing using multiple methodologies, and improved surgical transplantation and engraftment outcomes. Importantly, physiological parameters in rats are more similar to those in humans when compared with mice; these unique features of rats could be crucial for safety studies.

## 9. Conclusions

Although significant advances against human infectious diseases and cancers have been made over the past century, these diseases remain a significant source of health burden [142]. Several persistent infectious diseases, including HIV, continue to cause significant morbidity and mortality [142]. Additionally, despite novel therapies such as immunotherapies, cancers remain a major health burden [143]. HCMV-vectored vaccines can generate robust and persistent effector memory immune responses and provide a viable strategy for preventing and treating persistent infectious diseases and cancers [107]. Humanized mouse and rat models provide an excellent platform for accelerating the development of HCMV-based vaccines against persistent infectious diseases and cancers.

## Figures and Tables

**Figure 1 ijms-26-03082-f001:**
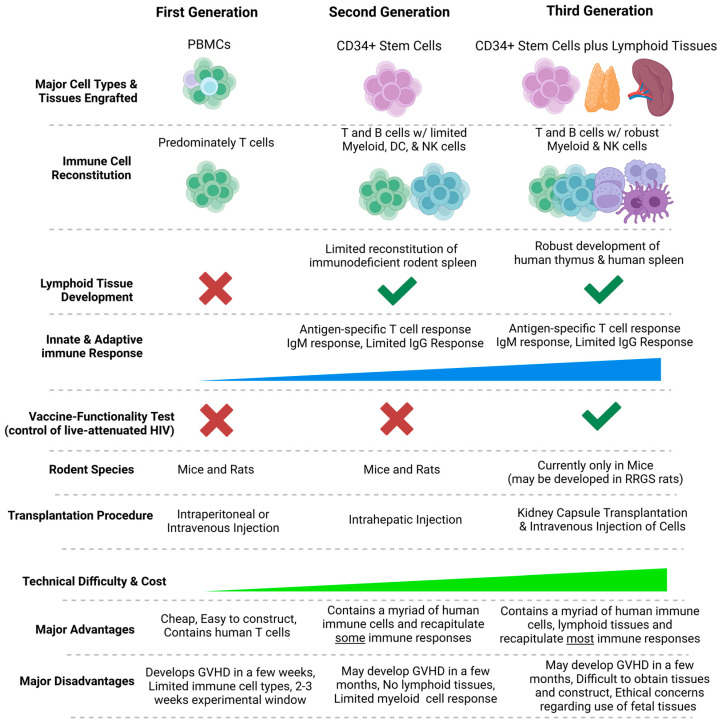
The generations of HIS rodent models. HIS-mouse models are broadly divided into three generations. First-generation HIS mouse models are reconstituted with PBMCs (or T cells from PBMCs), resulting in predominately T cell (CD4+ T cells) reconstitution in severely immunodeficient mice and rats. The major advantage of first-generation HIS mouse models is their low cost and ease of construction; however, they develop graft versus host diseases (GVHD) in a few weeks, resulting in a short experimental window. For second-generation HIS mouse models, immune reconstitution is mediated via transplantation of CD34+ hematopoietic stem cells. Various immune cell types are developed from CD34+ cells in immunodeficient mice and rats, albeit T and B cell reconstitution dominates, with limited reconstitution of myeloid and NK cells. The major advantage of second-generation HIS mouse models is their reconstitution with a myriad of human immune cells, albeit they lack human lymphoid tissues and associated myeloid cells and may develop graft versus host diseases (GVHD) in a few months. Third-generation HIS rodent models, including Bone Marrow-Liver-Thymus-Spleen (BLTS) and Bone Marrow-Liver-Thymus (BLT)-mice, are reconstituted with a myriad of immune cells via CD34+ stem cell transplantation and lymphoid tissues, and exhibit robust immune response. The robust development of the human thymus plus human spleen in third-generation HIS mice results in viremic control of live-attenuated HIV, which indicates a functional immune system. Advancements in immunodeficient rat models suggest that third-generation HIS rats could be generated. The major advantage of third-generation HIS mouse models is their reconstitution with a myriad of human immune cells and human lymphoid tissues, which can generate robust antigen-specific immune responses. However, third-generation HIS-mouse models may develop graft versus host diseases (GVHD) in a few months. Furthermore, third-generation HIS mouse models are difficult to construct, and ethical concerns exist regarding the use of fetal tissues.

**Figure 2 ijms-26-03082-f002:**
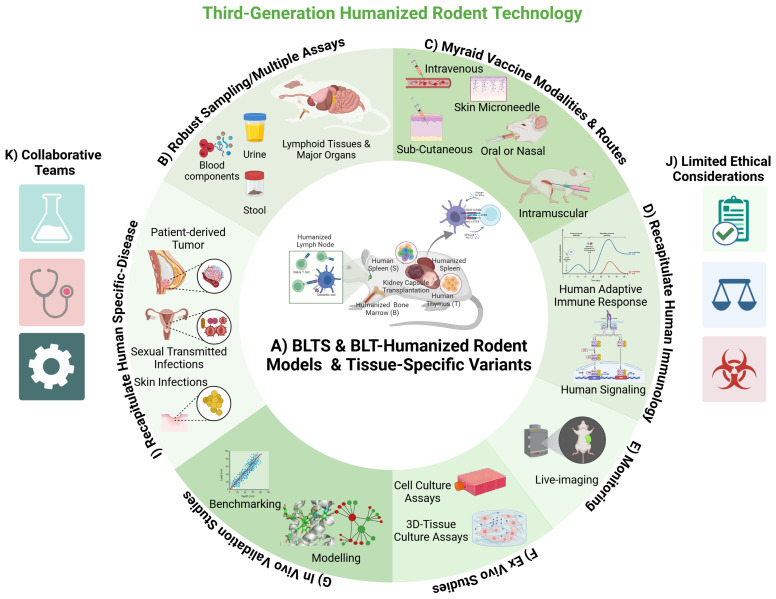
Schematic of third-generation HIS rodent models and their application in vaccinology research. (**A**) Third-generation HIS rodent models, including Bone Marrow-Liver-Thymus-Spleen (BLTS) and Bone Marrow-Liver-Thymus (BLT)-humanized mice, as well as tissue-specific variants (lung, skin, etc.) are reconstituted with a myriad of human immune cells and lymphoid tissues, and most accurately recapitulate the human immune system. The recent development of the Rat Rag−/− Gamma chain−/− human signal regulatory protein alpha-positive (RRGS) immunodeficient rat model will likely enable the development of third-generation HIS-rat models. (**B**–**G**,**I**) Third-generation HIS rodent models provide in vivo platforms for studying human-specific infectious diseases (such as HIV/AIDS and skin infections) and human tumor biology (including breast cancer), along with the associated immune responses to vaccines. HIS-rodent models allow comprehensive sampling of lymphoid tissues (spleen, thymus, lymph nodes), biological materials (blood, urine, stool), and major organs for investigating human diseases and vaccine-induced immune responses that cannot be addressed in clinical trials. Furthermore, these models enable the rapid evaluation of various vaccine modalities through multiple routes. A significant advantage of third-generation HIS-rodent models is their ability to replicate human-adaptive immune responses driven by human-specific molecular signaling. Rodent models also allow live monitoring of biological processes, enhancing mechanistic studies. Further ex vivo mechanistic studies can be conducted with tissues, cells, and biological materials from third-generation HIS rodent models to elucidate the mechanisms of human diseases and human immunity. Additional studies employing a systems biology approach in third-generation HIS rodent models could determine the efficacy, safety, and human immune correlates of novel vaccines against infectious diseases and cancers with a high degree of certainty. (**J**) The limited ethical, legal, and safety concerns involving rodent models make this technology ideal for biomedical research. (**K**) Studies in HIS rodent models are inherently collaborative and interdisciplinary, involving the application of concepts from molecular biologists and human samples (tissues and cells) and insights from clinicians and public health experts. Future efforts should include computational biologists to improve rigor in mechanistic studies in HIS rodent models.

**Figure 3 ijms-26-03082-f003:**
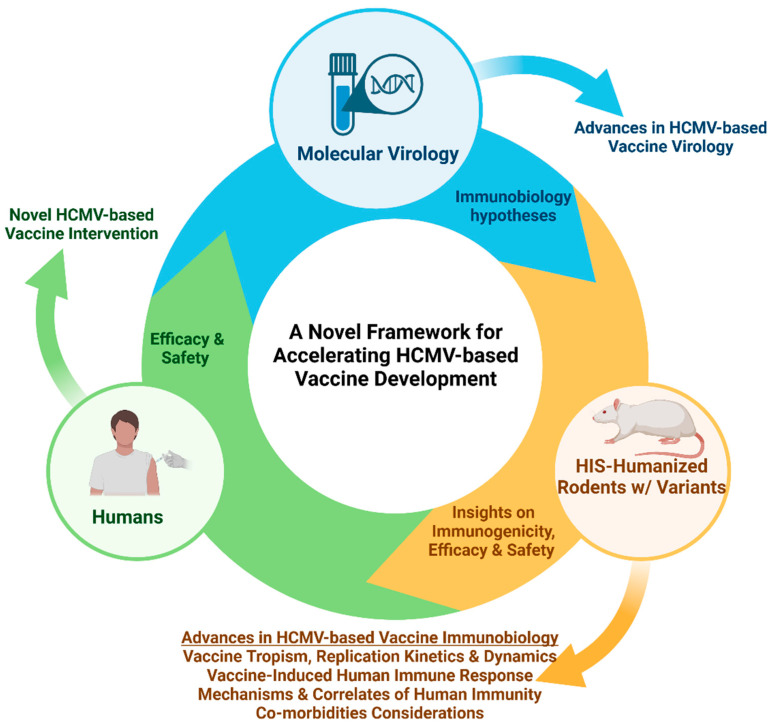
Schematic of a framework for accelerating HCMV-based vaccine development using HIS-rodent models. Advances in the molecular virology of HCMV-viral vector could spur novel immunobiology hypothesis that requires animal models with human tissues and cells. HIS rodent models (rats and mice) provide in vivo platforms for investigating the immunobiology of HCMV-based vaccines, which could enable rapid and rational implementation of clinical trials to determine efficacy and safety and subsequent approval.

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
