# Peer review of "Can Humanized Immune System Mouse and Rat Models Accelerate the Development of Cytomegalovirus-Based Vaccines Against Infectious Diseases and Cancers?"

_ijms, 2025, doi:10.3390/ijms26073082_

Round 1
Reviewer 1 Report
Comments and Suggestions for Authors
In this review paper, Craft et al. provide a detailed overview of the state of humanized mouse and rat models and their potential use in accelerating the development of vaccines using cytomegalovirus as a backbone. They also discuss their potential applications in cancer research.
There are some small changes that may improve the quality of this manuscript.
- In the development of humanized rodent models, several groups have significantly contributed to this area. Research teams from the US, Germany, and Japan have been involved in the creation and development of these models. Therefore, in addition to references 1 and 2, this reviewer suggests adding publications from these groups as well.
- Figure 1. Please add a line indicating the route of inoculation above the technical and difficulty This will help explain the complexity involved in the generation of these humanized rodents.
- Line 125-126. Please switch reference 2 to the original reference and apply this correction throughout the entire manuscript.
- Figure 2 is rather confusing. This reviewer suggests labeling it with three layers (a, b, and c) and referring to each layer when discussing an item of this graph in the text.
- Line 187. Please use original references.
- Lines 180-194. This paragraph discusses the use of humanized models in virology and refers to Figure 2. However, there is no mention of viruses in Figure 2.
- Lines 199, 212, 220, 227, 244, 253, 296, 325, 327, 328, 345, 350, 356, 468. Please add references.
- Line 247. This sentence starts with “several studies” however the reference is only Smith MS et al. Please add references or rephrase this sentence. In addition, please format this reference.
- Line 257. It mentions “several studies” but there are no references.
- Line 297. Please check reference 68. The sentence mentions the development of a rat model and the reference is a molecular biology technique.
- Line 378. Since the vaccine has tropism for human cells and what is reconstituted in these animals is the human immune system, instead of 'different organs,' it should be 'different immune system compartments”
- Line 429. Please check the name of the vaccine. HVTN 142 corresponds to the name of the phase 1 clinical trial no to the vaccine name.
- Line 474. Please use original reference.
- This manuscript would benefit from the addition of a conclusion paragraph. It is also missing the authors’ contributions and funding sections.
Author Response
Response to Reviewer #1
- In the development of humanized rodent models, several groups have significantly contributed to this area. Research teams from the US, Germany, and Japan have been involved in the creation and development of these models. Therefore, in addition to references 1 and 2, this reviewer suggests adding publications from these groups as well.
Response: We have added references on the significant of research groups from the US, Germany, and Japan to the development of HIS-rodent models.
- Figure 1. Please add a line indicating the route of inoculation above the technical and difficulty ……...
Response: We have added a line indicating the route of inoculation.
- Line 125-126. Please switch reference 2 to the original reference and apply this correction....
Response: We have added the original refences.
- Figure 2 is rather confusing. This reviewer suggests labeling it with three layers (a, b, and c) and referring to each layer when discussing an item of this graph in the text.
Response: We have labels to the layers for clarity.
- Line 187. Please use original references.
Response: We have added the original refences.
- Lines 180-194. This paragraph discusses the use of humanized models in virology and refers to Figure 2. However, there is no mention of viruses in Figure 2.
Response: We have removed the refence to figure 2.
- Lines 199, 212, 220, 227, 244, 253, 296, 325, 327, 328, 345, 350, 356, 468. Please add references.
Response: We have added refences.
- Line 247. This sentence starts with “several studies” however the reference is only Smith MS et al. Please add references or rephrase this sentence. Please format this reference.
Response: We have added more refences and revised the format of the Smith MS et al. refence.
- Line 257. It mentions “several studies” but there are no references.
Response: We have added refences.
- Line 297. Please check reference 68. The sentence mentions the development of a rat model and the reference is a molecular biology technique.
Response: We have removed that statement and the refence.
- Line 378. Since the vaccine has tropism for human cells and what is reconstituted in these animals is the human immune system, instead of 'different organs,' it should be 'different immune system compartments”
Response: We have added “'different immune system compartments”.
- Line 429. Please check the name of the vaccine. HVTN 142 corresponds to the name of the phase 1 clinical trial no to the vaccine name.
Response: We have corrected the statements, using VIR-1388 as the name for the vaccine and HVTN 142 as the name of the clinical trial.
- Line 474. Please use original reference.
Response: We have added the original refence.
- This manuscript would benefit from the addition of a conclusion paragraph. It is also missing the authors’ contributions and funding sections.
Response: We have added the authors’ contributions and funding sections.
Reviewer 2 Report
Comments and Suggestions for Authors
Dear authors: Congratulations on this excellent review. I have only minor comments/suggestions, which can be found in the attached document.

Author Response
Response to Reviewer #2
This manuscript explores the use of humanized immune system (HIS) rodent models in accelerating the development of human cytomegalovirus (HCMV)-based vaccines for infectious diseases and cancers. The authors provide a comprehensive review of HIS models, their advantages over traditional animal models, and their potential applications in vaccine research. Given the growing interest in alternative vaccine platforms, the discussion of HCMV-based vaccines is highly relevant. The figures depicting HIS-mouse models and their applications in vaccinology provide useful visual summaries of key concepts.
I agree with the authors regarding the relevance of evaluating the efficacy of and safety of HCMV-based HIV vaccines in HIS-rodents. On lines 468-471, the authors wrote: “Evaluating the efficacy and safety of HCMV-based HIV vaccines in HIS-rodents is worthwhile before embarking on clinical studies in humans to avoid catastrophic outcomes, such as the STEP HIV vaccine trial, which resulted in increased susceptibility to HIV transmission”.
- PLEASE ADD THE CORRESPONDING REFERENCE.
Response: We have added the reference.
- I RECOMMEND THE AUTHORS TO CAREFULLY CHECK THEIR DOCUMENT AS THERE ARE SOME IMPORTANT SENTENCES THAT ARE NOT REFERENCED. FOR EXAMPLE: on lines 476-479 THE AUTHORS WROTE:
“Similar concepts can also be applied to HCMV-based cancer vaccines, as recent
advances in tumor modeling in HIS-mice enable closing the translation gap between
laboratory research and clinical efficacy and safety studies”.
Response: We have added the references.
- I also recommend the authors to create a table including the main advantages and
disadvantages of the HIS models, as it is important to generate an objective, non-biased
report..
Response: We have added the main advantages and disadvantages of the HIS models in figure 1.